# Crosscutting environmental risk with design: A multi-site, multi-city socioecological approach for Iowa's diversifying small towns

**Benjamin Shirtcliff**[1]*, **Rosie Manzo**[1], **Rachel Scudder**[2]

**1** Landscape Architecture, Iowa State University, Ames, Iowa, United States of America, **2** Community and Regional Planning, Iowa State University, Ames, Iowa, United States of America

* bens@iastate.edu

## Abstract

Globally, the influx of refugee, migrant, and immigrant populations into small centers of industrialized agriculture has called attention to a looming public health crisis. As small towns shift from remote villages into rural, agri-industrial centers, they offer limited access to amenities needed to support human well-being. Our study focused on three Iowa towns that continue to experience an increase in under-represented minority populations and decline of majority populations as a proxy for studying shifting populations in an era of industrialized agriculture and global capital. We aimed to understand the socioecological impact of built environments—outdoor locations where people live and work—and likelihood of environmental exposures to impact vulnerable populations. Urban socioecological measures tend to present contradictory results in small towns due to their reliance on density and proximity. To compensate, we used post-occupancy evaluations (POE) to examine built environments for evidence of access to environmental design criteria to support healthy behaviors. The study systematically identified 44 locations on transects across three small towns to employ a 62 item POE and assess multiple environmental criteria to crosscut design with environmental health disparities. Principal-components factor analysis identified two distinct significant components for environmental risk and population vulnerability, supporting similar studies on parallel communities. Multilevel modeling found a divergence between supportive environmental design coupled with an increase environmental risk due to location. The combined effect likely contributes to environmental health disparities. The study provides a strategy for auditing small town built environments as well as insight into achieving equity.

## 1. Introduction: Small U.S. towns & rural built environments

Countries around the world are facing an increase in industrialized agriculture, growing urbanization, shifting populations, and a mounting public health dichotomy between urban/rural built environments [1, 2]. The current state of research on local life in small agrarian towns reveals a critical knowledge-gap linking built environments with health-promoting behaviors. Recently, the National Institutes of Health (NIH) modified health disparity populations to include people from underserved rural areas. NIH's change responds to the limited research in rural settings,

**Data Availability Statement:** All data used in this study has been made available as part of this submission.

**Funding:** Funding for this research was provided through grants from the National Institutes of Health LRP 2674-6 NIEHS; Center for Excellence in Arts and Humanities at Iowa State; Fieldstead & Co. Grant for Community Engagement; Honor's College funding from the Iowa State Foundation; and, a Project U-TuRN Mini-Grant through Iowa State's Presidential Interdisciplinary Research Initiative. All sources of funding were awarded to BS.

**Competing interests:** The authors have declared that no competing interest exist.

including measuring the impact of small-town environments on healthy behaviors [3, 4]. The intersection of environmental risk and design on vulnerable populations has received extensive study in urban settings but, as new U.S. policies suggest, rural areas and small towns pose a substantial gap in the literature. The study's objective is to correlate environmental risk with environmental design for evidence of factors potentially exacerbating health disparities faced by vulnerable populations; and, identify strategic opportunities for designers and planners.

## 1.1. Literature review

**1.1.1. Environmental justice and small town built environments.**   Globally, population shifts have led to an emergence of multiple health and well-being concerns, such as the mental health and physical health impacts of racism [5], an increase in cardiovascular disease among workers exposed to pesticides and fertilizers [6], an increase in asthma amongst young people [7–10], and physical and social isolation that prevents access to health-related services [11–13]. For example, as industrialized agriculture continues to develop, small towns have become gateways and job centers for a new, vulnerable workforce [1, 14, 15]. The situation is like what environmental justice advocates describe as a "double jeopardy" of injustice where people with the fewest resources reside in low-income communities with high level of environmental risk and unable defend against social threats like racism [9, 10, 16–18]. The recent change has left many decision makers wondering how to use limited community resources to address new and existing population needs [19–24]. In contrast to theories describing urban-rural change as evidence of environmental racism [2], our research paper seeks to understand how environmental risk and design in small towns may unintentionally impact vulnerable populations. Since social factors are unique for minority communities in rural environments, the paper will briefly discuss the environmental justice related to parallel communities.

**1.1.2. Parallel communities in rural Midwest.**   Small towns throughout the Midwest have experienced steady population loss, however towns that process farm products, provide farm labor, or employment in ag-related manufacturing replace decreasing numbers with a growing minority base—predominately first and/or second generation foreign-born, under-represented minority groups [25]. This attraction to small Midwestern started in the 1980's, following expanding meat processing and declining post-union salaries [26]. Despite population stabilization, Sandoval found vulnerabilities related to poor living conditions and *parallel communities*.

Parallel communities, populations that seldom interact due to work and shopping schedules, geography, and language barriers, can effectively destabilized local economies due to hidden "flows through employment recruitment networks, lending networks, remittance transfers, and smuggling networks" [27]. At one extreme, Sandoval, Nelson, and others found that individuals with questionable status due to the color of their skin, even residents with citizenship or green cards, felt compelled to isolate themselves because of their inferred 'illegal' identity [28]. Nelson et al., noted that the population shift has changed the ethnic composition of small towns, decreased tax-base, and created a form of residential segregation where *people live in parallel universes*, rarely interacting with one another [28]. In Postville, Iowa, for example, the local economy became dependent upon a workforce that was committed to investing somewhere else. The reliance on a parallel distribution of social, economic, and ecosystem resources eventually led to the collapse of Postville's economic and social systems, as will be discussed in the implications.

## 1.2. Justification of methods: Crosscutting environmental risk with design in three small towns in Iowa

To understand the physical reality of these parallel universes, the present study used a crosscutting approach which began with the use of transects (see below) to strategically cut across

small towns to relate environmental risk with design. Environmental risk coupled with poor environmental design has the capacity to impact everyone but likely to exacerbate vulnerability for under-represented and invisible populations. As suggested by Talen [29], due to historic patterns of uneven development and segregation, the present study included natural and infrastructural barriers, like rivers and highways, known to isolate neighborhoods. Following others, our study implemented multiple means to assess outdoor, built environments [30]. In the following sections, we operationalize built environments, review measurement approaches, and suggest transects to overcome the limitations of density-based models to study how environmental risks may impact daily stressors.

**1.2.1. The built environment and POE.**   The built environment is a commonly used term in the realm of public health, human sciences, and design. The Center for Disease Control (CDC) classifies the built environment as "all the physical parts of where we live and work." The broad reaching definition is focused in the current study to outdoor environments such as streetscapes, open spaces, social gathering spaces, and infrastructure [31]. Recent environmental health research has led to a myriad of urban studies that use population density as a proxy for built environments with lower levels of population density often showing equitable levels of access to green infrastructure [32]. Similarly, studies examining neighborhood characteristics have measured the built environment using features such as street tree density, residential density, intersection density, land-use mix, greenspace distribution, greenspace quality, and the walkability index that combines these features [30, 33, 34]. Small towns, with low population densities and lacking other forms of geographic density represent a paradox: the built environment is not comparable with common urban measures.

The rural-health paradox suggests that vulnerable and isolated populations in low-density areas appear to be less at-risk. Rural populations are less-likely to report symptoms because there are fewer individuals but also because of lack of access and awareness [35]. Similarly, in reference to commonly reported urban green and supportive benefits of nature [36], small, agrarian towns appear to be surrounded by green and nature (e.g., corn fields). However, industrialized agriculture has made these green landscapes volatile [9]. One similarity between urban and rural built environments is the threat of "natural" green spaces. Vacant lots in cities and rural, industrialized, green areas contain contaminated soils and plants [37]. In response, our study used a 62-item Likert scale to survey the perceivable quality of built environments through post-occupancy evaluation (POE) to measure the quality of environmental design to meet residential needs.

POE refers to auditing the quality and use of built environments after implementation and is predominantly used for buildings by architects and interior designers. Typically, trained observers score locations looking for issues like access, facilities, amenities, features, incivilities, safety, and usage. Scales such as the Natural Environment Scoring Tool or the Community Park Audit Tool provide a standardized means of assessing conditions of types of places for human populations [30, 38]. Our study employed a holistic model developed for outdoor residential environments using the validated instrument Cross and Küller developed in Sweden [39]. To validate this checklist, Cross and Küller "compared the experts' scores (r = .71) for each area to the satisfaction residents had regarding outdoor environment" (*ibid*., p. 79). The 62-item checklist (Table 1) has been used by others to study both *objective* measures of the built environment—explicitly measurable features like visibility of trash or water—and *subjective* measures—implicitly understood by trained professionals like legibility [40].

**1.2.2. Environmental risk, daily stressors, and environmental design.**   The impact of environmental risk is increasingly measured through daily stressors. Epidemiological research by Theall et al., has identified serious impacts of the built environment on stress-related inflammation and other health outcomes at an urban scale [see 41–43]. Such public health

**Table 1. POE items from cross's checklist.**

| Criteria | Measures | Number of Items |
|---|---|---|
| Physical Criteria | general layout, complexity & coherence, identity and affection, construction materials, greenery, climate, pollution | 37 |
| Social Criteria | meeting areas, privacy, security, traffic, and maintenance | 25 |

research provides precedents for the importance of measuring and classifying environmental risk in built environments [44]. Our study builds upon this area by incorporating environmental risk from pollution [9] and exposure to heat and wind, for example, that potentially exacerbate health-related problems [45]. While this study does not propose to measure individual-level stress, it does build upon a growing public health literature and key legislative documents such as the Clean Air Act and the Clean Water Act identifying the impact of unhealthy conditions on human well-being. Daily stressors found at the intersection of vulnerability and environmental risk include: *weathering* [46, 47], *individual risk* [8, 44, 48], *morbidity* [7, 9, 35, 49], and *human mortality* [6, 18, 50]. In response to environmental risk, vulnerable populations may benefit from environmental design that buffers stressors and provides a protective mechanism in neighborhood environments [51]. The following study looks at the intersection of environmental risk with design to formalize criteria planners and designers could assess in addressing public health disparities.

### 1.3. Research questions and hypothesis

The present study aims to understand how small towns could overcome parallel lives and provide environmental resources. The null hypothesis, built upon the environmental justice literature, is that vulnerable residents at the intersection of low-SES and minority status are just as likely as wealthier, educated, English-speaking residents to live and work in outdoor locations with supportive environmental design and that no difference would be found between small towns and average, urban populations. The study used a crosscutting, systematic sampling method through transects to identify whether vulnerable populations were at an increased risk for health disparities due to environmental risk and design.

## 2. Methods

The medium-sized, descriptive study of transect points across three small towns ($n = 44$) used a qualitative and quantitative comparative analysis approach to identify probabilistic relationships related to environmental risk and design [52]. The post-occupancy evaluation of study sites selected locations using a systematic sampling interval based upon variations in socio-ecological criteria, e.g. low-income housing in floodplains, through transects [53, 54]. Internal validity was achieved by evaluating each point twice in the field and once online by different pairs of trained graduate students. The modal score of each trice evaluated point was used to achieve reliability. Secondary data was obtained from publicly available reputable sources, Environmental Protection Agency, Center for Disease Control, and Census data. The study did not involve human subjects.

### 2.1. Study sites

Our study focuses on the built environments in three Iowan small towns deeply affected by the transnational shift in population and economic resources. Of Iowa's roughly three million

people, one-third, or one million live in small towns with a population below 10,000 [55]. The state average, accordingly, is considered mostly urban. The three towns, Perry, Ottumwa, and Marshalltown were selected based upon size, over 70 miles apart, exempt from a municipal statistical area, and documented population change.

## 2.2. Transects

In this study, each transect line had to cross environmental criteria potentially exacerbating risk, such as agricultural fields next to a public middle school, or support quality of life, like access to a park (see criteria and maps in Fig 1). Overall, four trained graduate student researchers from environmental design disciplines and the primary investigator identified multiple transects that cut across each small town. Transect lines were divided into equally spaced points with ½ mi. buffers and each point was field verified using the POE and secondary data for each point was downloaded from the EPA (Fig 1). Each transect conducted a post-occupancy evaluation (POE) of built environments using Cross's validated residential survey. The use of secondary environmental and vulnerability data provides insight into public health concerns and risks to human well-being. Finally, mapping and locating transect points, POE data, and secondary data, permitted spatial relationships to be analyzed.

## 2.3. Primary data collection and measures

Graduate research assistants—four across two years working in pairs—with extensive training in landscape architecture, architecture, and planning conducted the POE using the freely available, educational license of the mobile data collection platform Fulcrum (www.fulcrumapp. com). The mobile app geolocates survey points and functions through computers, smart phones, and through offline tablets if maps are downloaded prior to going into the field. For this study, an app was built using Cross's survey (see section 1.2.1), with the permission of the authors, that included the pictures and occasionally videos of each location. Research assistants assessed transect points using Cross's 4-point scale of agreement, for example "I very much agree that this area has no trash present" or "I mostly agree that this area contains a landmark that makes it easy to locate." The scale is written using negative in the question, so that "very much agree" scores will always overall rank in positive environmental characteristics. For example, a score of "Very Much Agree" that there are no traces of vandalism would be a positive measure even though the question is about vandalism. This permits a sum of all points and sums of point for each criterion to easily compare across locations. Final scores were numerically modified (1–4) to support statistical analysis. (Table 2).

## 2.4. Secondary data sources and measures

Transect points were triangulated with secondary data from the Environmental Justice Screening and Mapping tool (EJSCREEN) developed by the EPA (Table 3, S1 File). EJSCREEN is a publicly available data source that allows users to explore recent demographic and environmental indicators of environmental justice issues for specific geographic areas. Environmental indicators display potential sources of environmental pollutants and include eleven data points, that illustrate toxicity and proximity measures for air, waste, water, and soil. Demographic indicators focus on vulnerability and include six data points, including low-income, minority, less than high school education, linguistic isolation, and individuals under age five and over the age of 64. The use of multiple population vulnerability measures has been advocated by others as means of moving beyond poverty and/or SES as the primary predictor of environmental disparities [30, 56].

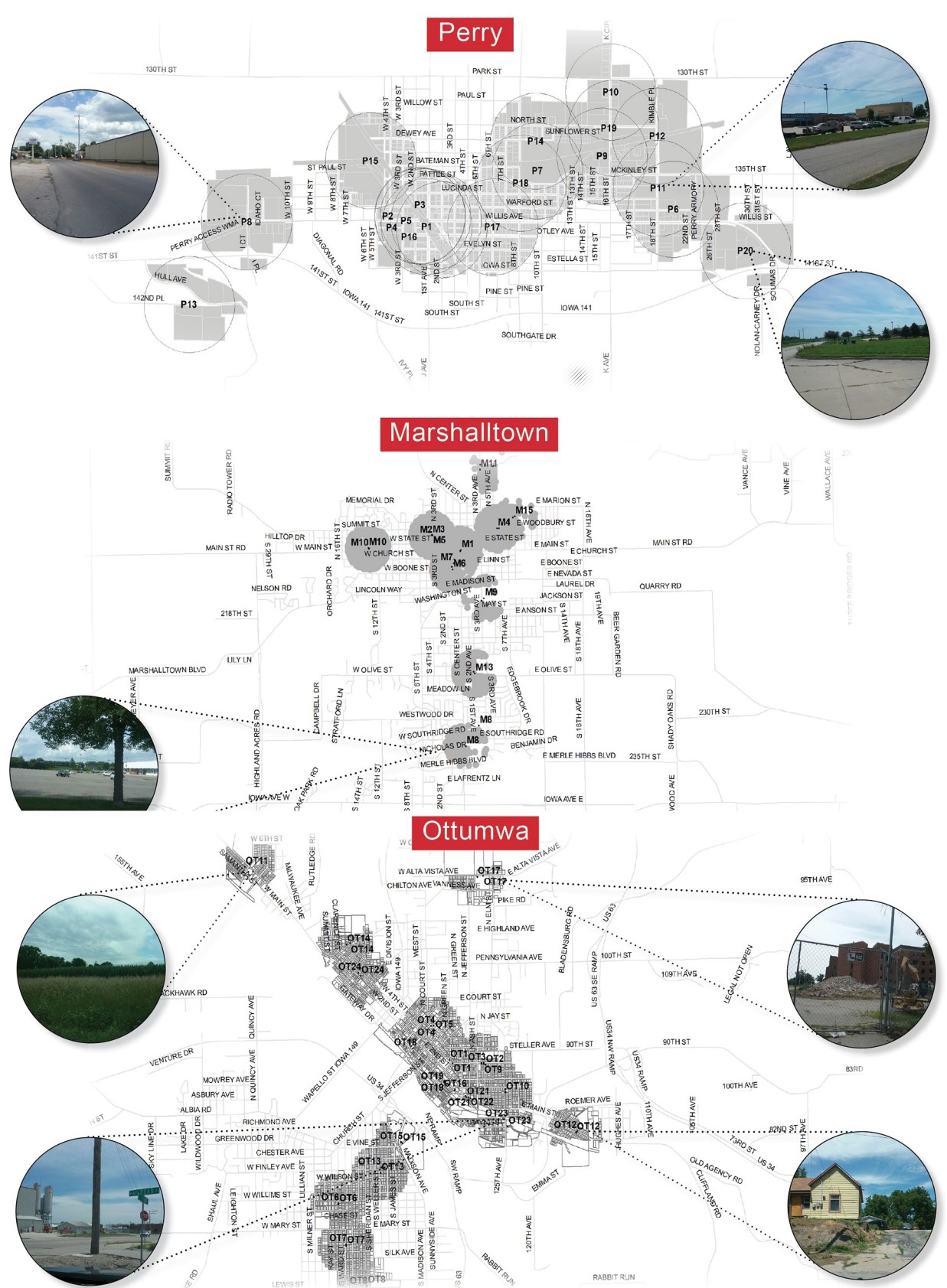

**Fig 1. Transects across Perry, Marshalltown, and Ottumwa follow the criteria: Includes at least one urban boundary, a natural edge, an agricultural edge, and an urban center; cross high and low elevation points, i.e. water's edge to hilltop or ridge and floodplain; pass across or close to at least one elementary, middle, or high school; includes a potentially harmful and a potentially beneficial infrastructure, like a canal, interstate, railroad, bike path, park, factory, or other major transportation infrastructure; includes multiple land use types: Residential, commercial, multifamily, industrial; be accessible at intervals by car, bike, or walking.** Transect buffers overlap in heterogeneous areas to capture increased complexity of built environments. Reprinted from Iowa DOT under Create Commons license of public domain data with a waiver of all rights including attribution CCZero license [2018].

## 2.5. Analyses

Data was cleaned and screened prior to analysis in SPSS 27. Analysis began by examining significant differences between transect points and an urban proxy, state averages, using t-tests, then constructed two principal component axes (PCA) to measure outcomes related to environmental risk and criteria related to vulnerability, and finally nested environmental exposures, vulnerability, and POE surveyed environmental design variables were entered in multilevel models to interpret what environmental design features were most likely to explain intersections of risk and design among vulnerable populations. The study was designed to use multilevel modeling to cluster variables by location (nesting dependent and independent variables into transect points) to look for within and between location effects. Overall, 206 variables were collected across 44 transect points (five points were eliminated at an early stage in the data collection process due to miscommunication). We used t-tests (Table 3) to see if environmental risk and vulnerability differed in small towns when compared to state population averages—state averages were used as a proxy for urban environments as EPA averages reflect population, which in Iowa is 70% urban. This approach is in concert with the rural-health paradox by Kim et. al. noted above. Similar to Rigolon et al. [57], we entered environmental risk as an outcome variable using a principal-component analysis (PCA). We entered vulnerability as a criterion variable using the same process. PCA converts potentially correlated variables

**Table 2. Variables Collected through Fulcrum show the number of transect points, the average group criteria score for each transect point across the three cities, and the average scores across all cities.**

| Average Transect Point Scores and Average Sum of Scores for Each Category per City | | | | | | | | | | |
|---|---|---|---|---|---|---|---|---|---|---|
| Category | Perry | | | Ottumwa | | | Marshalltown | | | Overall | |
| | n | AVG (SD) | AVG (SD) | n | AVG (SD) | Avg Sum (SD) | n | AVG (SD) | Avg Sum (SD) | AVG (SD) | Avg Sum (SD) |
| **Physical Criteria (# of items)** | | | | | | | | | | | |
| General Layout (6) | 20 | 1.95 (.57) | 11.70 (3.40) | 14 | 2.42 (.47) | 14.50 (2.82) | 10 | 1.60 (.44) | 9.60 (2.63) | 2.02 (.59) | 12.11 (3.52) |
| Complexity and Coherence (5) | 20 | 2.71 (.70) | 11.65 (3.60) | 14 | 2.34 (.70) | 9.36 (2.80) | 10 | 2.58 (.63) | 8.90 (3.38) | 2.56 (.69) | 10.30 (3.47) |
| Identification and Affection (8) | 20 | 2.55 (.54) | 19.50 (4.57) | 14 | 2.50 (.70) | 14.78 (3.53) | 10 | 2.64 (.71) | 14.20 (7.70) | 2.55 (.54) | 16.77 (5.62) |
| Construction Materials (5) | 20 | 1.98 (.48) | 9.75 (2.61) | 14 | 1.59 (.29) | 7.93 (1.44) | 8 | 1.93 (.92) | 6.63 (2.89) | 1.84 (.56) | 8.55 (2.61) |
| Greenery (4) | 20 | 2.01 (.80) | 6.50 (3.82) | 14 | 2.24 (.61) | 7.86 (2.96) | 7 | 2.11 (.71) | 5.86 (4.88) | 2.11 (.71) | 6.85 (3.73) |
| Climate (2) | 20 | 2.38 (.86) | 4.75 (1.71) | 14 | 1.68 (.61) | 3.36 (1.22) | 10 | 2.20 (.75) | 4.40 (1.51) | 2.11 (.81) | 4.23 (1.61) |
| Pollution and Noise (2) | 20 | 2.18 (1.04) | 4.35 (2.08) | 14 | 1.2 (.38) | 2.43 (.76) | 10 | 1.75 (.54) | 3.50 (1.08) | 1.77 (.87) | 3.55 (1.74) |
| Ecological Sustainability (5) | 20 | 3.51 (.40) | 13.83 (3.73) | 13 | 2.33 (.55) | 4.46 (1.8) | 6 | 3.15 (.80) | 8.83 (7.52) | 3.04 (.75) | 9.73 (5.84) |
| **Social Criteria** | | | | | | | | | | | |
| Place (8) | 20 | 2.65 (.55) | 20.40 (5.21) | 14 | 3.14 (.48) | 23.71 (4.25) | 10 | 2.52 (.82) | 15.7 (7.01) | 2.77 (.64) | 20.39 (6.04) |
| Privacy 5) | 20 | 2.76 (.77) | 12.35 (3.42) | 14 | 2.40 (.63) | 10.36 (3.25) | 9 | 2.68 (.51) | 9.56 (2.70) | 2.63 (.68) | 11.12 (3.38) |
| Security and Traffic Control (8) | 20 | 2.41 (.40) | 17.5 (2.97) | 14 | 2.60 (.48) | 18.07 (3.75) | 10 | 2.36 (.38) | 15.6 (3.66) | 2.46 (.42) | 17.25 (3.44) |
| Maintenance (5) | 20 | 1.38 (.67) | 5.15 (2.89) | 14 | 1.58 (.65) | 5.14 (2.14) | 10 | 1.43 (.94) | 3.70 (.83) | 1.43 (.72) | 4.82 (2.37) |

Each group has multiple measures of evaluation, 2–8, as indicated in the parentheses, so the mode of each item was averaged into its group criteria. Average sums illustrate the highest negative value, i.e. I strongly disagree that this area is well organized (4), so the 6-item general layout could have a range from 6 of meets environmental design criteria to 24 of severely lacking.

**Table 3. Variables downloaded using EPA's EJscreen for each 1/2 mi. transect point.**

| Category | Selected Variables | *n* | Range, Average (SD) | State Avg. | Sig. |
|---|---|---|---|---|---|
| Environmental | Particulate Matter (PM 2.5 in ug/m3) | 44 | 8.67–9.47, 9.04 (.32) | 9.23 | ~ |
| Environmental | Ozone (ppb) | 44 | 39.8–40.3, 39.9 (.2) | 40.5 | ~ |
| Environmental | NATA Diesel PM (ug/m3) | 44 | .28–2.22, .71 (.34) | 0.586 | $t = 2.5, p < .05$ |
| Environmental | NATA Air Toxics Cancer Risk (risk per MM) | 39 | 23–41, 31.87 (3.78) | 30 | $t = 3.09, p < .01$ |
| Environmental | NATA Respiratory Hazard Index | 39 | .66–2.0, 1.14 (.23) | 1.1 | ~ |
| Environmental | Traffic Proximity and Volume | 39 | 2.4–3500, 743 (954) | 1500 | ~ |
| Environmental | Lead Paint Indicator | 39 | .19–.94, .62 (.18) | 0.42 | $t = 7.2, p < .01$ |
| Environmental | Superfund Proximity | 39 | .02-.03, .021 (.005) | 0.098 | ~ |
| Environmental | RMP Proximity | 39 | .26–3.6, 1.53 (.84) | 1.2 | $t = 2.4, p < .05$ |
| Environmental | Hazardous Waste Proximity | 39 | .01–1.2, .24 (.36) | 0.53 | ~ |
| Environmental | Wastewater Discharge Indicators | 39 | .00-.16, .023 (.035) | 0.018 | ~ |
| Demographic | Demographic Index | 39 | 10%-62%, 39% (12%) | 21% | $t = 20.5, p < .01$ |
| Demographic | Minority Population | 39 | 3%-62%, 33% (16%) | 13% | $t = 12.7, p < .01$ |
| Demographic | Low Income Population | 39 | 15%-73%, 46% (13%) | 30% | $t = 21.5, P < .01$ |
| Demographic | Linguistically Isolated Population | 39 | 0%-19%, 7% (5.4%) | 2% | $t = 8.2, p < .01$ |
| Demographic | Population with Less Than High School Education | 39 | 4%-34%, 19% (6.5%) | 8% | $t = 18.2, p < 01$ |
| Demographic | Population under Age 5 | 39 | 1%-16%, 7.75% (3.9%) | 6% | $t = 12.4, p < .01$ |
| Demographic | Population over Age 64 | 39 | 6%-25%, 14% (5.3%) | 16% | $t = 16.3, p < .01$ |
| Demographic | Population | 39 | 10–1348, 442 (392) | | |

Variables also show significant deviation from state averages. As forementioned, the state average population is urban permitting state averages to be used as proxy. All data has been deidentified and uploaded as part of this manuscript.

from observations into linearly uncorrelated composite values. PCA is used for exploratory data analysis and for predictive modeling. Finally, to assess criteria from the physical environment, we compared both environmental risk and vulnerability scales to the 62-items checklist through a series of multilevel models.

Multilevel modeling (MLM) was used to examine environmental design criteria across sites as they relate to the PCA environmental risk outcome and vulnerability criteria. This strategy permits a more reliable means of calculating the similarities of differences (i.e., residuals) within and between sites. Following Hoffman [58], restricted maximum likelihood (REML) was used to make estimates and inferences about covariance parameters. First, an initial unconditional model free of any predictors was used to measure the amount of variation in environmental risk, differentiating between-site variance from within-site variance. MLM does not violate the assumption of independent observations when modeling nested data, thereby permitting a more accurate, real-world assessment. Upon setting up the model, we measured the intraclass correlation coefficient (ICC)—a key statistic that is commonly used to evaluate similarities for several "classes" in a school. The ICC measures how well residuals are correlated and can be used to indicate the degree to which observations taken at different locations are stable within each site. Of primary conceptual interest, a high ICC indicates that observations are reliable indicators of differences between locations.

## 3. Results

### 3.1. Urban/Rural dichotomy, vulnerability and environmental risk

In response to the question whether people in small towns were at higher risk of exposure when compared to their urban counterparts, environmental risk from pollutants was not found to be

equal across the three small towns in our study and, in some cases, these towns evidence significantly higher risks of exposure than state averages (Table 3). Small town exposure to diesel ranges from .28 to 2.22 with a state average of .59, air toxics recognized to increase risk of cancer range from 23–41 with a state average of 30, lead paint from older homes .19 to .94 (.42 average), and proximity to potential chemical accidents range .26–3.6 with a state average of 1.2.

Above average environmental risk can remain unnoticed in healthy adults but can pose a serious threat to vulnerable populations. The demographic index, the EPA's scale of poverty and minority status, ranges in the three study towns from 10% - 62% with a state average of 21%, minority 3% - 62%, low-income 15%-73%, linguistic isolation 0% - 19%, less than high school 4% - 34%, under age 5 ranges 1% - 16%, and over age 64 ranges 6% - 25%. All the social vulnerability indexes were significantly higher than state averages across the three small towns.

### 3.2. Outcome variable: Environmental risk

Environmental Risk criteria from the EPA's EJscreen (Appendix 1 in S1 File) were extracted to reveal two components: the first had an eigenvalue of 2.2 and accounted for 43.8% of variance across 44 transect locations. The second component is orthogonal to the first factor and had an eigenvalue of 1.3 and accounted for 26% of the variance of the 5 variables: PM 2.5, NATA Diesel PM, NATA Air Toxics Cancer Risk, Lead Paint, and RMP Proximity. The risk factors positively loaded onto the first factor, with especially high factor loadings for Air Toxics, Particulate Matter and Lead Paint indicators (Fig 2); this first factor is thus collectively termed

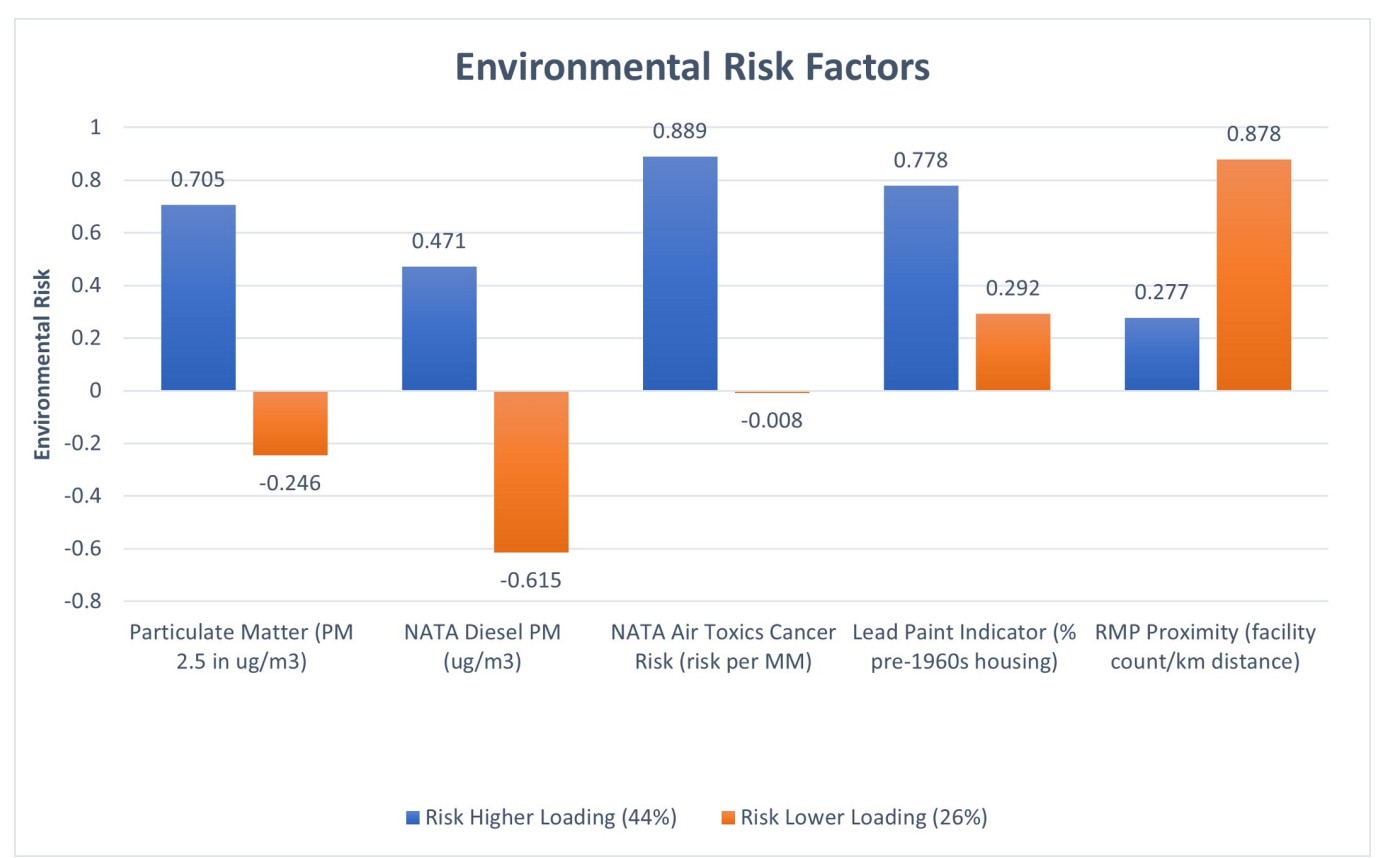

**Fig 2. Environmental Risk Factors indicates the environmental factors across the transects likely to represent the greatest combined risk.** Higher scores indicate higher risk of environmental exposure.

Environmental Risk. The Environmental Risk variable was normally distributed (Appendix 2 in S1 File) with lower scores representing decreased risk and higher scores increased risk of exposure. The second component suggests some environmental risk is driven by high RMP proximity and low NATA diesel PM values, suggesting this second factor is influenced by small town proximity to RMP facilities and rural isolation. The study used the first component, Environmental Risk, as the outcome variable for further analysis.

### 3.3. Criterion variable: Social vulnerability

Social vulnerability was measured through a (PCA) to create a factor score that merged demographic variables (Minority Population, Low Income Population, Linguistically Isolated Population, Population with Less Than High School, Population under Age 5, and Population over Age 64) into a single construct for vulnerability. The PCA revealed two components. The high vulnerability variable was extracted from the first component with an eigenvalue of 3.52 that explained 58.7% of the total variance in six of seven variables: minority population, low-income, linguistic isolation, less than high school educated, and presence of children under 5 (Fig 3). The second component had a low an eigenvalue of 1.008 that explained 16.8% of the variance, predominately low-income and over 64 (see Appendix 3 in S1 File). This vulnerability variable was normally distributed.

### 3.4. Environmental design: Measures of physical and social conditions

The eight physical and four social criteria measures from post-occupancy evaluations were transformed from categorical variables indicating level of agreement (very much agree (1) to

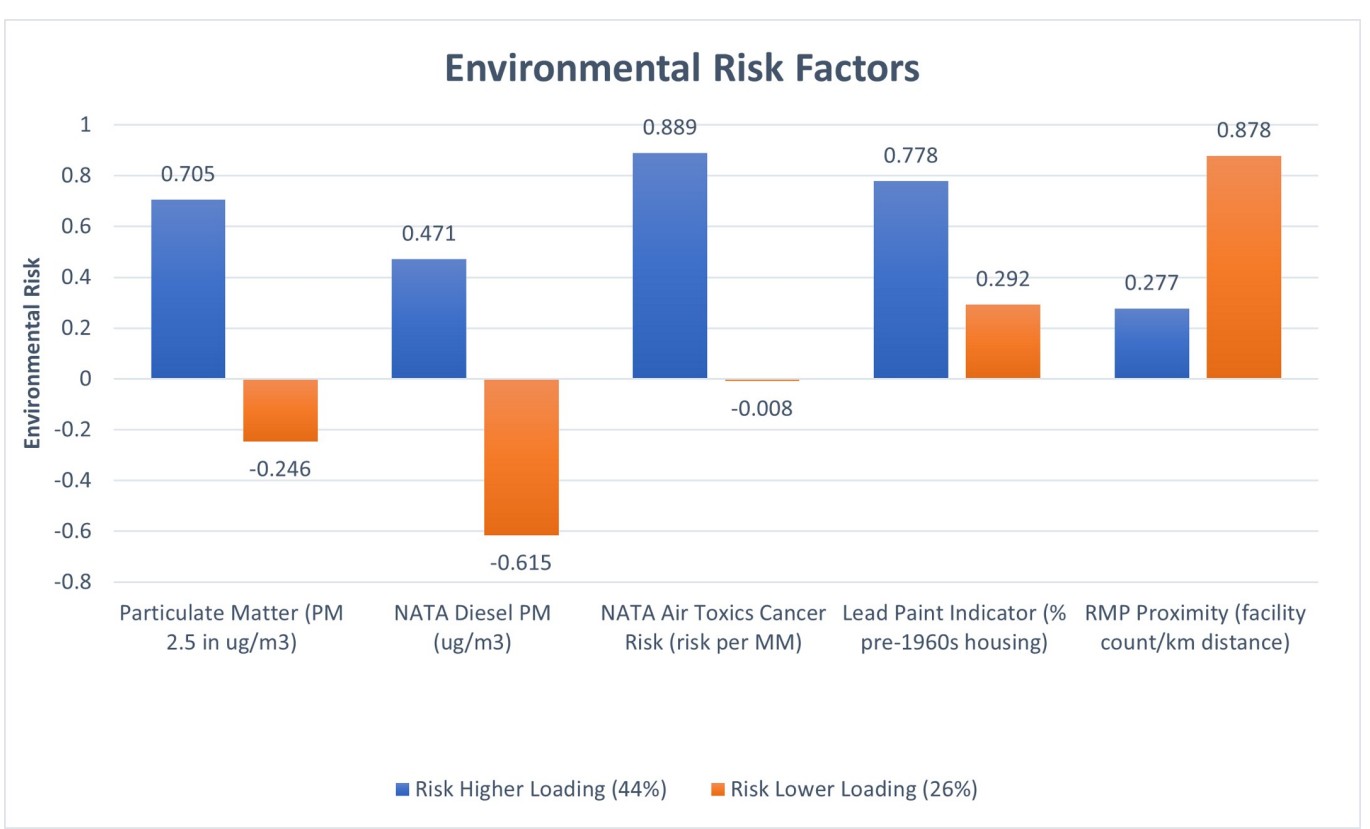

**Fig 3. Population vulnerability factors identify the two components from PCA and statistically represent the parallel populations within each small town.** Higher scores indicate more vulnerability.

very much disagree (4)); and, then the multiple indicators for each transect point were averaged per group and summed to represent each group total score. Overall physical (.32) and social (.65) criteria showed a significant correlation (p < .05, see Appendix 4 in S1 File) with demographic index; areas with minorities showed a positive correlation (.40) with place; low income inversely with ecological sustainability (-.51) and pollution (-.46); population under 5 inversely with identification (-.35), place (-.38), and physical overall (-.39)—vulnerable populations tend to be associated with unsupportive, residential environments. Like vulnerability and environmental risk, a PCA was used to explore how environments rated with poor supportive qualities related to vulnerability and environmental risk. The outcome Environmental Risk variable had a significant correlation with the residential environments PCA (.49), p < .001, suggesting that environments with high levels of environmental risk were likely to also score poorly in terms of environmental design.

### 3.5. POE and MLM

Next, the analyses examined whether the transect location explained variance and therefore served as an important indicator of exposure to environmental risk. Using multilevel modeling, we used the intraclass correlation, (ICC = .55 (.22), p < .05), to demonstrate that 55% of environmental risk in small towns is related to specific locations. The significant ICC indicates that the relative exposure of populations to environmental burdens is not randomly distributed in small towns. Demonstrating a significant ICC is important for justifying further analysis into what environmental design factors may account for further increase in environmental inequity.

Similar to other neighborhood effect studies, a series of multilinear models were run to see what physical environmental factors were most likely to load variance onto the outcome variable environmental risk [51, 59]. Following Gerring's [52] suggestion for studies with medium-sized samples, variables from the post-occupancy evaluation survey were dichotomized—scores 1 and 2 were transformed to 0 and scores 3 and 4 were transformed to 1—and dummy coded as 0, agree that the built environment positively supports the variable, and 1, disagree. Data was grouped by subjects using the 44 location points with the outcome variable of environmental risk. The environmental design variables (62) from Cross and Kuller [39, 60] were individually entered—this is due to the inherent limitations in degrees of freedom of 44 transect points—as factors according to their group, like general layout, complexity and character, identity, etc. (see Table 4). Coefficients overall suggest an increase associated with environmental risk in environments without mystery, complexity, history, water, materiality, big trees, shield, biodiversity, and enclosure (Fig 4). Access indicated a significant inverse effect, suggesting that places with more traffic have increased environmental risk. Each of these variables maintained significance when controlling for increases in population vulnerability.

## 4. Discussion

This study sought out to address how small towns, struggling with a decline in economic resources and the emergence of parallel lives, might prioritize meaningful investments into the built environment for vulnerable populations. The study replicates findings supporting the urban/rural dichotomy with multiple environmental exposures higher in small towns than state population averages. Moreso, the study also found that within small towns, vulnerable populations were more likely to be in locations with higher levels of environmental risk, potentially increasing daily stressors. Daily stressors relate to the distribution of environmental exposures and may affect chronic inflammation that directly impacts human well-being

**Table 4. Fixed effects from 62 item POE across 44 transect points.**

| POE Var. | Effect | Coefficients (β) | Standard Error | Approx. df | t Ratio | P (2-sided) | 95% Confidence Interval | |
|---|---|---|---|---|---|---|---|---|
| | | | | | | | Lower | Upper |
| Access | | | | | | | | |
| | High Access* | 0.69 | 0.31 | 37 | 2.21 | 0.03 | 0.06 | 1.32 |
| | Low Access* | -0.21 | 0.36 | 37 | -2.52 | 0.02 | -1.62 | -0.18 |
| Mystery | | | | | | | | |
| | High Mystery | -0.30 | 0.21 | 37 | -1.42 | 0.16 | -0.74 | 0.13 |
| | Low Mystery* | 0.63 | 0.31 | 37 | 2.03 | 0.05 | 0.00 | 1.25 |
| Complexity | | | | | | | | |
| | High Complexity* | -1.06 | 0.47 | 37 | -2.25 | 0.03 | -2.02 | -0.11 |
| | Low Complexity* | 0.12 | 0.50 | 37 | 2.38 | 0.02 | 0.18 | 2.19 |
| History | | | | | | | | |
| | High History* | 0.18 | 0.33 | 37 | -2.13 | 0.04 | -1.38 | -0.03 |
| | Low History* | 0.89 | 0.37 | 37 | 2.38 | 0.02 | 0.13 | 1.65 |
| Water | | | | | | | | |
| | High Water* | 0.73 | 0.17 | 37 | -2.38 | 0.02 | -0.75 | -0.06 |
| | Low Water** | 1.13 | 0.28 | 37 | 3.98 | 0.00 | 0.55 | 1.70 |
| Materiality | | | | | | | | |
| | High Materiality* | 0.13 | 0.42 | 37 | -2.14 | 0.04 | -1.77 | -0.05 |
| | Low Material* | 1.04 | 0.45 | 37 | 2.29 | 0.03 | 0.12 | 1.96 |
| Big Trees | | | | | | | | |
| | High Trees | 0.20 | 0.32 | 37 | -1.95 | 0.06 | -1.26 | 0.02 |
| | Low Trees* | 0.70 | 0.36 | 37 | 2.22 | 0.03 | 0.07 | 1.54 |
| Shield | | | | | | | | |
| | High Shield | 0.19 | 0.30 | 37 | -1.82 | 0.08 | -1.16 | 0.06 |
| | Low Shield* | 0.74 | 0.35 | 37 | 2.11 | 0.04 | 0.03 | 1.45 |
| Biodiversity | | | | | | | | |
| | High Biodiversity** | 0.27 | 0.29 | 37 | -3.16 | 0.00 | -1.51 | -0.33 |
| | Low Biodiversity** | 1.19 | 0.33 | 37 | 3.61 | 0.00 | 0.52 | 1.87 |
| Enclosure | | | | | | | | |
| | High Enclosure | 0.27 | 0.25 | 37 | -1.71 | 0.10 | -0.92 | 0.08 |
| | Low Enclosure * | 0.69 | 0.31 | 37 | 2.18 | 0.04 | 0.05 | 1.32 |

* Dependent Variable Environmental Risk. Intercept is positively rated environment and named effect is negatively rated environment, so higher coefficients indicate increase risk associated with lack of main effect, like shield. Aside from Access—the environmental criteria is deemed as a positive in Cross's survey but access includes proximity to traffic and vehicles—, all variables show an increase in environmental risk associated with poor environmental criteria with significant effects (p < .05) italicized. Significance noted at the * = p < .05; ** = p < .01, and *** = p < .001 level.

(Fig 5). A closer examination of the built environment revealed that environmental design correlates with parallel lives and identifies multiple opportunities that could be changed to improve equity in environmental design. Study findings suggest pathways for small towns struggling with the socio-economic resources needed to respond to an emerging crisis.

Our study followed a socioecological approach using transects to permit a cross-sectional, multi-site approach to analyze a holistic socio-ecological picture of small town, built- environments. The approach aimed to overcome limitations of density or political boundaries that often reveal contradictory results, like identifying the benefits of living close to a park [36] or the harms of going to school where farmers spray pesticides [9]. The use of transects enables

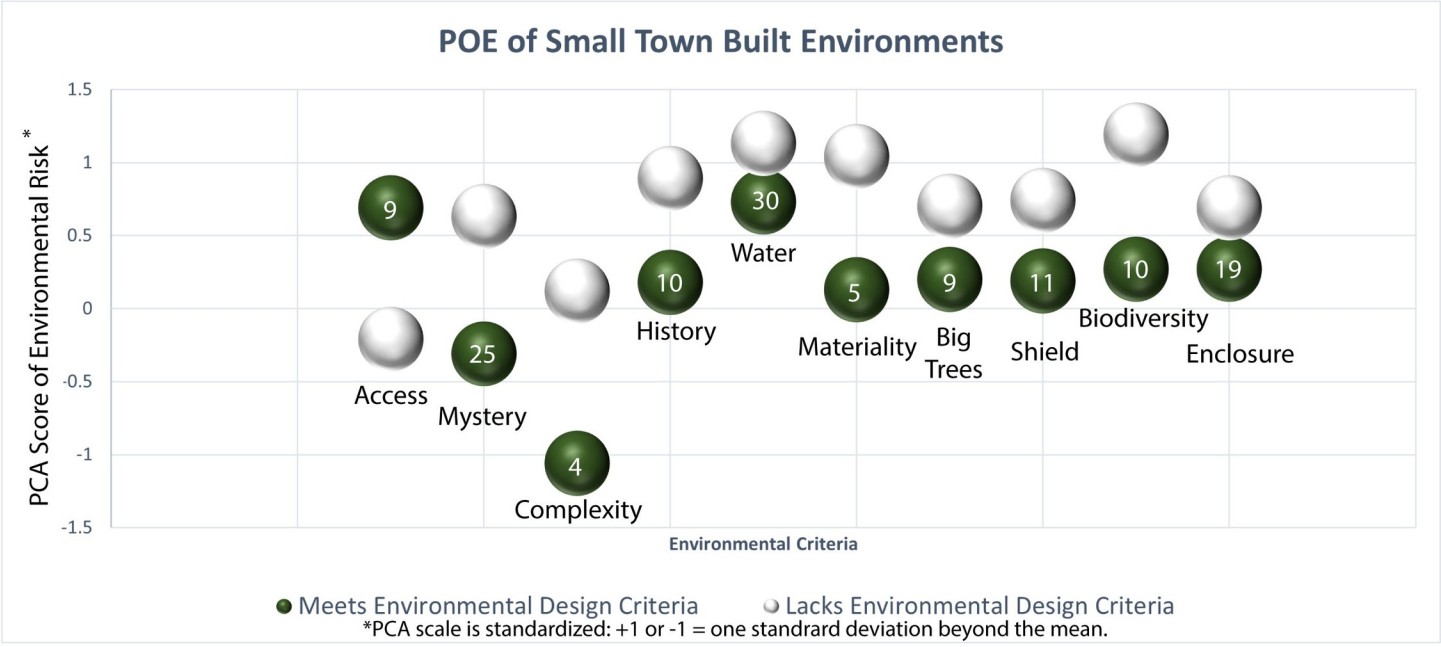

**Fig 4. Coefficients from a series of multilevel models identify built environment criteria differentiated as lacking environmental design criteria also being associated with our PCA scale of increase in levels of environmental risk—standardized so that a score of 1 us one standard deviation above the mean—and higher quality environments with lower levels of environmental risk.** See Appendix 6 in *S1 File* for a matrix of exploratory Global Moran I's post-hoc clusters analyzed in GeoDA 1.18.0. Locations consistently cluster from high quality and low risk to low quality and higher environmental risk. Note that environmental design measures should not be compared in terms of lacking or possessing environmental design criteria—one location may or may not possess multiple criteria—, instead this indicates how specific criteria, like biodiversity, related to environmental risk nested within each location. The number inside each dark green circle indicates the frequency of meeting environmental design criteria.

higher specificity of environmental exposures, vulnerabilities, and access to environmental benefits, to demonstrate inequalities in existing built environments and prioritize planning and urban design efforts.

**The Challenge of Establishing a Causal Health Pathway for Environmental Justice and Built Environments**

HISTORIC INEQUALITIES DUE TO:

Individual Experience

Space

Group Identity

Security Threats related to: environment, displacement, family, housing, water, employment, food...

Environmental Justice is Inequality of the Built Environment to Impact Health

Weathering
Risk
Morbidity
Mortality

Stress Induced Inflammatory Response

Chronic Exposures

Acute Exposures

Difficult, impossible, or inappropriate to change

Daily Stressors that affect individual health

Distribution of Environmental Benefits and Burdens

Acute and Chronic Impacts on Individual Health

**Fig 5. A non-causal model prioritizing improvement to built environments to counter deteriorating conditions and buffer vulnerable populations.**

Our post-occupancy evaluation provided a means of understanding environmental aspects that could support a design-response. The physical criteria of general layout, complexity and coherence, identity, greenery, habitat, and the social characteristic of privacy were significantly lacking in transect points with increased vulnerability and environmental risk. Privacy, for example, is a substantial threat for populations that have been identified as needing to be invisible to protect their livelihoods. Landscape architects, planners, and designers have the tools to improve environmental design by selecting the correct tree species, screening, and vegetation and by working with vulnerable communities to establish gardens with visual interest that enhance privacy. Space, individual experience, and group identity are difficult or impossible to change, however the built environment provides multiple alternatives that may diminish the impact of daily stressors on human weathering, risk, morbidity, and mortality (Fig 5).

The study is in common ground with similar studies pushing for landscape science, planning, and design to go beyond finding causal impacts on human health. The complex relationship of population vulnerability and environmental risk are potentially exacerbated by environmental stressors that lead to weathering, risk, morbidity, and mortality (See Fig 5). Instead, researchers, policymakers, and practitioners hoping to address current and burgeoning public health crises could rely on the availability of reliable data and present capabilities to respond within improved environmental design. The approach builds upon the precautionary principle [61] by asking how do we in the face of uncertainty address increasing risk by linking science, ethics, and practice?

### 4.1. Implications

Earlier the paper introduced the trouble with parallel rural lives and their demise about Postville, Iowa. In 2008, Postville experienced one of the largest Immigration and Customs Enforcement (ICE) worksite raids in the US [14]. "These small Midwestern towns, no longer tranquil, are now nodes within the global industrial network of food production, a network teeming with immigration-related issues such as the unauthorized status of many workers, exploitation of workers, and new and often "invisible" human, gender, and racial dynamics" (Sandoval, 2013, p. 181). The raid had immediate impacts on increased criminal behavior, long-term impacts on human health, and solidified the insecurity for underrepresented minorities working and living in small towns [47, 62]. Although the influx of foreign-born workers and their families to small towns has enabled economic growth in the hands of a local few, the stability of small towns is fragile. A decline in local investment coupled with aging infrastructure is likely to impact the built environments in small towns, potentially compounding deleterious effects as vulnerable populations bring families and become established.

### 4.2. Limitations

Landscape architects and professional planners are trained to observe environmental characteristics to improve human well-being. The use of Cross's Professional Residential Survey provided a validated instrument helping to assess what small towns could do to help improve built environments for vulnerable populations living within environments of risk of exposure. The survey captured criteria relevant to the elements of the built environment that professional designers and planners can address. A limitation of the approach is that the survey tool was created for designed residential settings and does not capture the heterogenous nature of development found within small towns. Small towns, for example, typically lack a planning office or a set of design guidelines, and, instead, address land use issues on an as needed basis. A future study could benefit from this paper by adjusting Cross's survey to account for the somewhat haphazard nature of small-town development. The survey instrument provided

relevant information needed to assess how well residential settings supported human well-being. However, due to study limitations related to time and funding, healthy behaviors were not measured. Although each location was surveyed three times, the study is limited through the use of implicit or subjective, design-related measures to study place effects. Other limitations include the use of secondary data for measuring environmental risk and population vulnerability; potential sampling bias since study sites were selected using a systematic sampling interval based upon variations in socio-ecological criteria; and, focusing on three small towns that all underwent diversification. The use of passive samplers to measure air quality, including communities that remain homogenous, and community surveys to ask residents about perceived environmental risks would contribute to further grounding research in this area.

## 4.3. Significance

The study makes a significant contribution to a growing area of research on disproportionate burdens vulnerable populations face regarding environmental benefits and burdens. Lack of access to green space has been identified as an environmental injustice by several researchers [30, 32, 38, 57, 63], however spatial models continue to rely on density and political boundaries to infer environmental justice. Such models are known to report misleading and contradictory findings in small towns.

As Breslow indicates, justice (along with security, resilience, and sustainability) is a crosscutting category that pulls from multiple aspects of the built environment, specifically: capabilities, conditions, and connections [64]. An environmental justice model that stops at comparing SES and access to green oversimplifies complex built environments and socioecological conditions on human health outcomes. Our non-causal model (Fig 5) proposes a means of contextualizing environmental justice within a built environment framework for human well-being. Our study goes beyond spatial autocorrelations to demonstrate paths to improve equity by crosscutting environmental risk from spatial data with field-measured environmental design.

## 5. Conclusion

Small towns throughout the Midwest began diversifying in the 1980's and simultaneously witnessed a decline in their economic tax-based as higher income earners relocated to major urban areas. Small towns evidence multiple characteristics described by urban design and planning researchers as the key ingredient to successful, walkable, urban environments, e.g., New Urbanism. The structure is clearly in place; however, history differentiates in how new populations are directly impacted by supportive built environments. Shifting, vulnerable populations—as characterized by underrepresented minority status in once all-Caucasian communities, linguistic isolation, below high-school education, age under 5 and over 64—are more likely to live in conditions that currently may not adequately support human well-being and are more likely to experience environmental risk.

Design activists can achieve environmental justice goals by impacting health effects (chronic and acute) that directly relate to the mortality, morbidity, risk, and weathering of vulnerable populations (Fig 5). Our study suggests that the professional practices like landscape architecture who are responsible for the management, planning, and design of the land should play a ubiquitous role in how daily stressors are translated into individual outcomes. First, we must accept that individual experience, group identity, and space play a fundamental role in daily stressors. Chronic stressors related to environmental exposures and acute stressors related to visibility and access to supportive space can be mediated through supportive environmental design. Landscape architecture prides itself on major parks, i.e., the High-line, and

environmental remediation, i.e., Fresh Kills, but seems to continue to neglect the necessity of the banal, everyday "human environment" where a sidewalk, street tree, and crosswalk make a fundamental difference. While this research is in an early phase, findings suggest that small towns could counter a mounting global public health crisis with low-cost interventions.

## Supporting information

**S1 File. Appendices.**
(DOCX)

**S1 Data.**
(XLSX)

## Acknowledgments

The project benefited greatly from interdisciplinary collaborators in Iowa State's Department of Kinesiology, Dr. Meyer, and Dr. Ellingson, and Extension, Dr. Wolseth and Dr. Seeger. The project was completed through the efforts of multiple undergraduate Honor's students, and the following graduate researchers: Kwadwo Gyan, Eric Lawrence, and Mahsa Adib. The paper benefited from multiple reviewers who added clarity and substantial quality to the organization of text, legibility of figures, and interpretation of the results.

## Author Contributions

**Data curation:** Rachel Scudder.

**Formal analysis:** Benjamin Shirtcliff, Rachel Scudder.

**Funding acquisition:** Benjamin Shirtcliff.

**Investigation:** Benjamin Shirtcliff, Rosie Manzo.

**Methodology:** Benjamin Shirtcliff.

**Project administration:** Benjamin Shirtcliff, Rosie Manzo.

**Software:** Rachel Scudder.

**Supervision:** Benjamin Shirtcliff.

**Validation:** Benjamin Shirtcliff, Rachel Scudder.

**Visualization:** Rachel Scudder.

**Writing – original draft:** Benjamin Shirtcliff, Rosie Manzo.

**Writing – review & editing:** Benjamin Shirtcliff.

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
