## [Decision Letter · Decision Letter 0]

8 Jul 2020

PONE-D-20-06092

“Ground Truthing” Environmental Barriers to Human Well-Being: Translational Research using a Multi-Site, Multi-City Approach for Iowa’s Diversifying Small Towns

PLOS ONE

Dear Dr. Shirtcliff,

Thank you for submitting your manuscript to PLOS ONE. After careful consideration, we feel that it has merit but does not fully meet PLOS ONE’s publication criteria as it currently stands. Therefore, we invite you to submit a revised version of the manuscript that addresses the points raised during the review process.

We look forward to receiving your revised manuscript.

Kind regards,

Tzai-Hung Wen, Ph.D.

Academic Editor

PLOS ONE

Journal Requirements:

3. We note that Figure 1 in your submission contain map images which may be copyrighted. All PLOS content is published under the Creative Commons Attribution License (CC BY 4.0), which means that the manuscript, images, and Supporting Information files will be freely available online, and any third party is permitted to access, download, copy, distribute, and use these materials in any way, even commercially, with proper attribution. For these reasons, we cannot publish previously copyrighted maps or satellite images created using proprietary data, such as Google software (Google Maps, Street View, and Earth). For more information, see our copyright guidelines: http://journals.plos.org/plosone/s/licenses-and-copyright.

3.1.    You may seek permission from the original copyright holder of Figure 1 to publish the content specifically under the CC BY 4.0 license.

3.2.    If you are unable to obtain permission from the original copyright holder to publish these figures under the CC BY 4.0 license or if the copyright holder’s requirements are incompatible with the CC BY 4.0 license, please either i) remove the figure or ii) supply a replacement figure that complies with the CC BY 4.0 license. Please check copyright information on all replacement figures and update the figure caption with source information. If applicable, please specify in the figure caption text when a figure is similar but not identical to the original image and is therefore for illustrative purposes only.

Reviewers' comments:

Reviewer's Responses to Questions

**Comments to the Author**

1. Is the manuscript technically sound, and do the data support the conclusions?

Reviewer #1: Yes

Reviewer #2: Partly

2. Has the statistical analysis been performed appropriately and rigorously? 

Reviewer #1: Yes

Reviewer #2: No

3. Have the authors made all data underlying the findings in their manuscript fully available?

Reviewer #1: Yes

Reviewer #2: No

4. Is the manuscript presented in an intelligible fashion and written in standard English?

Reviewer #1: Yes

Reviewer #2: No

5. Review Comments to the Author

Reviewer #1: This study involves an interesting analysis of the built environment and correlates with well-being, specifically in rural areas with changing demographics. Analysis of the data appears sound overall, and the paper is generally well written. I have only a few comments and suggestions to improve the clarity of the presentation.

The term “ground-truthing” may be a bit misleading. The study uses both data derived from existing geospatial data sets and on-the-ground observations, but there is no verification of the geospatial data using the observations, as ground-truthing would imply.

I am somewhat familiar with POE for buildings, not for outdoor spaces. If this is an adaptation, or novel aspect of this work, please explain.

By line 295 (and Table 3), it became apparent that the comparison is mainly between small towns and state averages, and not between rural areas and diversifying rural areas. Some of the introduction and background (and the title of the paper) led me to expect an analysis of the effects of diversification, but I suppose that would require a longitudinal (long-term) study, or a broader study of small towns including some that have diversified and others that have not.

Specific Comments:

Line 50: Spell out SES the first time used.

Line 64: Briefly define “parallel lives.”

Line 130: It’s not clear if the study is considering actual physical barriers (highways, railways, waterways) that can promote segregation of populations.

Lines 370-374: A figure (bar chart) or table might be helpful for presenting these results.

Lines 398-400: It may be helpful to define some of these variables, unless they are well known in the field of landscape architecture.

Reviewer #2: This manuscript intended to study the equity of environmental exposure and its built environment factors of the exposure in three small towns in Iowa. Specifically, the authors aimed to compare the exposure between small towns and general urban populations. To evaluate environmental exposure, the secondary EPA data were collected and compared with a state average. Then a post-occupancy evaluation was conducted to explore built environment characteristics. This is an interesting topic and it’s good to see this counterbalance aside from a myriad of urban studies on this topic. However, this manuscript is not ready to be published due to the fact that the analysis and rigorous in writing are not of sufficient quality. So I recommend major revision for this manuscript. My main concerns are listed below.

1. The comparison between small towns and the state average is biased (Table 3), because the transect points are selective and supposed to have higher exposure to pollutants compared to the average in town – “each transect line had to cross environmental criteria potentially exacerbating risk, such as agricultural fields next to a public middle school” (line 229)

2. The aim of this research currently stands is blurry given to the chosen terminology “environmental barrier”. In the literature in environmental health this term is often referred to the barrier for disabled people, or barriers to outdoor activities. However, this study is focusing on “exposure to environmental pollutants”. Plus the point about environmental exposure is also not clear in the introduction. It is highly recommended to make the research goal clearer and coherent throughout the title, abstract, as well as the introduction.

(1) in the abstract, the aim of study is described in a very general way: “We aimed to understand the impact of built environments—all the physical parts of where people live and work—to contribute to healthy behaviors for vulnerable populations.” Please be specific on which part of the built-environment. I am also wondering that did the authors include any variables of “healthy behaviors” in this work? Why is environmental exposure missing in this objective?

(2) In section 1.3 research question, the single hypothesis is that the environmental exposure in the small towns is not different from the exposure that general urban populations hold. Here, the role of built environment is missing in the hypothesis, which is conflicting to point (1).

(3) following the previous point, if the equality of environmental exposure is the only research question, table 3 will be the only relevant result. Why POE analysis, PCA and MLM analysis are needed?

3. It is not justified the design of multilevel model. If the class here is the towns, there would be only three groups. The suggested group size for multilevel model is 50 (Moineddin 2007). Otherwise, for a sample size of 39, a simple map could be clear enough to see the variation between transect locations if that is the only purpose to use multilevel model.

Here are some minor comments.

1. Replace the comma by a full stop in the end of the abstract.

2. Line 59 it’s unclear that if Ozkan is a kind of evaluation or an author name of the cited work until checking the reference.

3. Line 115 what does the combination “[emphasis added] (2008)” mean?

4. Please revisit some sentences and paragraphs. For instance,

(1) can you put the concrete meaning of environmental barriers in line 129 to the first glance around line 52?

(2) Lines 144-146 the authors argue that traditional built-environment metrics such as density are not applicable to rural environments. Right after, can you state what BE components you use for this study instead?

(3) Line 202 “The built environment in small towns may serve parallel populations in a dichotomous manner with underrepresented minorities encountering a disproportionate access to environmental harms and benefits” and following with the hypothesis “The hypothesis, …, is that vulnerable residents at the intersection of low-SES and minority status are just as likely as wealthier, educated, English-speaking residents to live in environments lacking supportive socio- and natural ecosystem services and that no difference would be found…”. What is the connection between these two sentences? Why the provided context in the first sentence drives such an opposite hypothesis? Would you consider rephrase?

(4) Line 212 methods, it is not clear that what the (n=39) sample size means without reading the text afterward. Does it mean towns? Residents? Transects?

(5) Line 267 how those secondary station data fits into transect points? What's the interpolation approach? How many points of observations were calculated around the study sites?

5. At the paragraph lines 136-151, the authors did not mention components of the social environment, making the social criteria in Table 1 come at a surprise.

6. Please check if the POE variables in Table 6 and Table 2 are consistent. Can you put the definition of each variable in method (e.g. unique and history sound very fuzzy)?

7. The authors need to describe the population characteristics of those who conducted POE and discuss the limitation when generalize their perceptions.

8. The data sources of lead pain index and RMP facilities are missing.

6. PLOS authors have the option to publish the peer review history of their article (what does this mean?). If published, this will include your full peer review and any attached files.

Reviewer #1: No

Reviewer #2: No

---

## [Author Response · Author response to Decision Letter 0]

5 Oct 2020

Response to Reviewers: this includes reviewer comments. The word version uploaded might be easier to read since it includes track changes and all of my responses are in blue.

3. We note that Figure 1 in your submission contain map images which may be copyrighted. All PLOS content is published under the Creative Commons Attribution License (CC BY 4.0), which means that the manuscript, images, and Supporting Information files will be freely available online, and any third party is permitted to access, download, copy, distribute, and use these materials in any way, even commercially, with proper attribution. For these reasons, we cannot publish previously copyrighted maps or satellite images created using proprietary data, such as Google software (Google Maps, Street View, and Earth). For more information, see our copyright guidelines: http://journals.plos.org/plosone/s/licenses-and-copyright.

RESPONSE:

The base map roads are from the Iowa Department of Transportation and fall within the Creative Commons CCZero license of public domain data with a waiver of all rights including attribution. All other map material was created by the author. 

3.1. You may seek permission from the original copyright holder of Figure 1 to publish the content specifically under the CC BY 4.0 license.

RESPONSE: Added note to figure caption of CCZero license.

3.2. If you are unable to obtain permission from the original copyright holder to publish these figures under the CC BY 4.0 license or if the copyright holder’s requirements are incompatible with the CC BY 4.0 license, please either i) remove the figure or ii) supply a replacement figure that complies with the CC BY 4.0 license. Please check copyright information on all replacement figures and update the figure caption with source information. If applicable, please specify in the figure caption text when a figure is similar but not identical to the original image and is therefore for illustrative purposes only.

Reviewers' comments:

Reviewer's Responses to Questions

Comments to the Author

1. Is the manuscript technically sound, and do the data support the conclusions?

Reviewer #1: Yes

Reviewer #2: Partly

2. Has the statistical analysis been performed appropriately and rigorously? 

Reviewer #1: Yes

Reviewer #2: No

3. Have the authors made all data underlying the findings in their manuscript fully available?

Reviewer #1: Yes

Reviewer #2: No

4. Is the manuscript presented in an intelligible fashion and written in standard English?

Reviewer #1: Yes

Reviewer #2: No

5. Review Comments to the Author

Reviewer #1: This study involves an interesting analysis of the built environment and correlates with well-being, specifically in rural areas with changing demographics. Analysis of the data appears sound overall, and the paper is generally well written. I have only a few comments and suggestions to improve the clarity of the presentation.

The term “ground-truthing” may be a bit misleading. The study uses both data derived from existing geospatial data sets and on-the-ground observations, but there is no verification of the geospatial data using the observations, as ground-truthing would imply.

Good suggestion. It would be inappropriate to suggest that geospatial environmental risk data could be “ground truthed” to predict poor POE. The objective was to see if there was a double-threat of environmental risk and poor, built environmental design. Title changed to reflect the intent of the paper: “Ground-truthing environmental risk with design”. Also added clarification of ground truthing in the methods section to due with visible verification of existing conditions. 

I am somewhat familiar with POE for buildings, not for outdoor spaces. If this is an adaptation, or novel aspect of this work, please explain.

Revised the methods section to clarify that the POE used an outdoor, residential tool developed by researchers in Sweden. 

By line 295 (and Table 3), it became apparent that the comparison is mainly between small towns and state averages, and not between rural areas and diversifying rural areas. Some of the introduction and background (and the title of the paper) led me to expect an analysis of the effects of diversification, but I suppose that would require a longitudinal (long-term) study, or a broader study of small towns including some that have diversified and others that have not.

This is an excellent point. State averages in the EPA data reflect urban environments because the majority of the population is urban, whereas rural areas remain outliers because they don’t have the density. I revised this section to note that state averages were used as a proxy for urban populations, noting that rural communities deviate substantially. Also, a comparison study of small towns that have diversified and those that have not is an excellent idea! I’ve added that to the limitations. I hope the inclusion of these revisions meets the reviewer’s expectations. 

Specific Comments:

Line 50: Spell out SES the first time used. Addressed.

Line 64: Briefly define “parallel lives.” Addressed in lines 49 – 52. 

Line 130: It’s not clear if the study is considering actual physical barriers (highways, railways, waterways) that can promote segregation of populations. Added the sentence: As suggested by Talen [36], the present study included natural and infrastructural barriers, like rivers and highways, known to isolate and segregate communities (see transects). 

Lines 370-374: A figure (bar chart) or table might be helpful for presenting these results. Environmental risk and population vulnerability are now figures as bar charts. The visual comparison of the divergence between the two factors helps to tell the story, thanks for the suggestion.

Lines 398-400: It may be helpful to define some of these variables, unless they are well known in the field of landscape architecture. Good suggestion but it might be a challenge to respond within word limits. The concepts are well known in the landscape architecture community. To help I included a reference here to the manuscripts by Cross and Kuller that go into detail on what each of these concepts mean. 

Reviewer #2: This manuscript intended to study the equity of environmental exposure and its built environment factors of the exposure in three small towns in Iowa. Specifically, the authors aimed to compare the exposure between small towns and general urban populations. To evaluate environmental exposure, the secondary EPA data were collected and compared with a state average. Then a post-occupancy evaluation was conducted to explore built environment characteristics. This is an interesting topic and it’s good to see this counterbalance aside from a myriad of urban studies on this topic. However, this manuscript is not ready to be published due to the fact that the analysis and rigorous in writing are not of sufficient quality. So I recommend major revision for this manuscript. My main concerns are listed below. RESPONSE: noted the myriad of urban studies papers on this topic. The original contribution from this research is its focus on small towns, which remain relatively understudied in this area. 

1. The comparison between small towns and the state average is biased (Table 3), because the transect points are selective and supposed to have higher exposure to pollutants compared to the average in town – “each transect line had to cross environmental criteria potentially exacerbating risk, such as agricultural fields next to a public middle school” (line 229) RESPONSE: The first paragraph of the methods section acknowledges this approach, “The post-occupancy evaluation of study sites selected locations using a systematic sampling interval based upon variations in socio-ecological criteria, e.g. low-income housing in floodplains, through transects.” The approach overcomes known political ecology constraints, like red-lining or census tracts, that have either negatively impacted communities through segregation or don’t represent communities due to low-density. In response to the reviewer comment, I have added sampling bias to the limitations section to further acknowledge the approach. 

2. The aim of this research currently stands is blurry given to the chosen terminology “environmental barrier”. In the literature in environmental health this term is often referred to the barrier for disabled people, or barriers to outdoor activities. However, this study is focusing on “exposure to environmental pollutants”. Plus the point about environmental exposure is also not clear in the introduction. It is highly recommended to make the research goal clearer and coherent throughout the title, abstract, as well as the introduction. RESPONSE: This is a really important comment and we appreciate the feedback. Please see paper for full extent of revisions, as the comment demanded a major revision. A summary of the changes follows: the title has been changed to emphasize the relationship between environmental risk and design with similar changes throughout the paper; environmental barrier removed and changed to environmental exposure and environmental design to be more specific. The abstract was revised substantially based upon this feedback. Hopefully, the objective is clearer and the use of POE to measure environmental support for behaviors is clearer.

(1) in the abstract, the aim of study is described in a very general way: “We aimed to understand the impact of built environments—all the physical parts of where people live and work—to contribute to healthy behaviors for vulnerable populations.” Please be specific on which part of the built-environment. RESPONSE: changed physical parts to outdoor environments. This eliminates vagueness related to housing, workplaces, and other indoor or otherwise inaccessible locations for survey research. 

I am also wondering that did the authors include any variables of “healthy behaviors” in this work? RESPONSE: Added a sentence that healthy behaviors were not measured during the course of the study and emphasized environmental design to support healthy behaviors. Due to funding limitations, the survey portion of the research was eliminated, so no human subjects research was conducted. The POE and MLM strategy serves as a proxy to examine the extent to which the built environment supports activities related to healthy behaviors, which is a common approach in ground truthing research methods. 

Why is environmental exposure missing in this objective? RESPONSE: revised the abstract to clearly indicate the study is looking at the correlation of environmental risk and design. 

(2) In section 1.3 research question, the single hypothesis is that the environmental exposure in the small towns is not different from the exposure that general urban populations hold. Here, the role of built environment is missing in the hypothesis, which is conflicting to point (1). RESPONSE: removed “socio- and natural ecosystem services” and replaced it with supportive environmental design. 

(3) following the previous point, if the equality of environmental exposure is the only research question, table 3 will be the only relevant result. Why POE analysis, PCA and MLM analysis are needed? RESPONSE: see notes throughout. This point has helped bring clarity throughout the paper.

3. It is not justified the design of multilevel model. If the class here is the towns, there would be only three groups. The suggested group size for multilevel model is 50 (Moineddin 2007). Otherwise, for a sample size of 39, a simple map could be clear enough to see the variation between transect locations if that is the only purpose to use multilevel model. RESPONSE: Excellent point although it might be difficult to map 206 variables across 39 locations or to statistically demonstrate significant differences. As noted, MLM has traditionally been used for students in classrooms in schools, but needing to achieve a threshold of 50 is a new constraint and one that I am not familiar. Neither Tabachnick (2007) or Hoffman (2007) suggest a group size of 50 as a threshold. Theall et al., referenced in this paper, found MLM produces reliable statistical analyses with group sizes <5. In justification of the method for the paper, the ICC and Betas from the MLM (Figure 5) demonstrate the environmental design specific to locations with supportive and non-supportive environments exist in the nested transect points across small towns. Parallel environments exist and they correlate with environmental risk. I added a better description of the MLM process and why this particular statistic was used because the study was designed to work with this statistical approach.

Here are some minor comments.

1. Replace the comma by a full stop in the end of the abstract. Revised.

2. Line 59 it’s unclear that if Ozkan is a kind of evaluation or an author name of the cited work until checking the reference. Added date.

3. Line 115 what does the combination “[emphasis added] (2008)” mean? Changed emphasis to italics.

4. Please revisit some sentences and paragraphs. For instance,

(1) can you put the concrete meaning of environmental barriers in line 129 to the first glance around line 52? Revised. 

(2) Lines 144-146 the authors argue that traditional built-environment metrics such as density are not applicable to rural environments. Right after, can you state what BE components you use for this study instead? Revised. 

(3) Line 202 “The built environment in small towns may serve parallel populations in a dichotomous manner with underrepresented minorities encountering a disproportionate access to environmental harms and benefits” and following with the hypothesis “The hypothesis, …, is that vulnerable residents at the intersection of low-SES and minority status are just as likely as wealthier, educated, English-speaking residents to live in environments lacking supportive socio- and natural ecosystem services and that no difference would be found…”. What is the connection between these two sentences? Why the provided context in the first sentence drives such an opposite hypothesis? Would you consider rephrase? Entire paragraph revised to make this clearer. Thank you for the suggestion. 

(4) Line 212 methods, it is not clear that what the (n=39) sample size means without reading the text afterward. Does it mean towns? Residents? Transects? Revised. 

(5) Line 267 how those secondary station data fits into transect points? What's the interpolation approach? How many points of observations were calculated around the study sites? This section was confusing and has been revised to be clearer. 

5. At the paragraph lines 136-151, the authors did not mention components of the social environment, making the social criteria in Table 1 come at a surprise. Revised. 

6. Please check if the POE variables in Table 6 and Table 2 are consistent. Can you put the definition of each variable in method (e.g. unique and history sound very fuzzy)? Table 6 uses some of the variables from table 2 but only the ones that were significant. The other reviewer also requested these terms to be defined but it would greatly expand the length of this already lengthy paper to explain. The reference to Cross and Kuller who created the scale and have published twice on the items has been emphasized. 

7. The authors need to describe the population characteristics of those who conducted POE and discuss the limitation when generalize their perceptions. Added description of the graduate students with extensive training in design disciplines to be fitting of a POE. 

8. The data sources of lead pain index and RMP facilities are missing. Removed text from document and redirected to appendices which provide a thorough list of sources. 

General response to reviewer two. Thank you for your thorough feedback. The paper clearly suffered from too much jargon and we’ve thoroughly vetted the paper seeking to add clarity. Probably the most important point was that socio- ecosystem resources didn’t appear in the hypothesis as what was measured. The revised paper eliminates this over-complicated phrasing and focusing on environmental design to make it clear that POE makes sense. 

6. PLOS authors have the option to publish the peer review history of their article (what does this mean?). If published, this will include your full peer review and any attached files.

Do you want your identity to be public for this peer review? For information about this choice, including consent withdrawal, please see our Privacy Policy.

Reviewer #1: No

Reviewer #2: No

---

## [Decision Letter · Decision Letter 1]

16 Dec 2020

PONE-D-20-06092R1

Ground Truthing” Environmental Risk with Design: Translational Research using a Multi-Site, Multi-City Approach for Iowa’s Diversifying Small Towns

PLOS ONE

Dear Dr. Shirtcliff,

Thank you for submitting your manuscript to PLOS ONE. After careful consideration, we feel that it has merit but does not fully meet PLOS ONE’s publication criteria as it currently stands. Therefore, we invite you to submit a revised version of the manuscript that addresses the points raised during the review process.

We look forward to receiving your revised manuscript.

Kind regards,

Tzai-Hung Wen, Ph.D.

Academic Editor

PLOS ONE

Reviewers' comments:

Reviewer's Responses to Questions

**Comments to the Author**

1. If the authors have adequately addressed your comments raised in a previous round of review and you feel that this manuscript is now acceptable for publication, you may indicate that here to bypass the “Comments to the Author” section, enter your conflict of interest statement in the “Confidential to Editor” section, and submit your "Accept" recommendation.

Reviewer #3: (No Response)

Reviewer #4: All comments have been addressed

2. Is the manuscript technically sound, and do the data support the conclusions?

Reviewer #3: Partly

Reviewer #4: Yes

3. Has the statistical analysis been performed appropriately and rigorously? 

Reviewer #3: I Don't Know

Reviewer #4: Yes

4. Have the authors made all data underlying the findings in their manuscript fully available?

Reviewer #3: Yes

Reviewer #4: Yes

5. Is the manuscript presented in an intelligible fashion and written in standard English?

Reviewer #3: No

Reviewer #4: Yes

6. Review Comments to the Author

Reviewer #3: This manuscript presented a quantitative analysis for the understanding of equity of environment conditions and well being in small towns or rural areas in Iowa, USA. The topic is important and interesting. But, the manuscript is not ready for publication. The current status of the manuscript was poorly written, with too much of missing information, e.g. the data collection was not clear, how the data was analyzed from one step to another was not clearly described, and the most critical issue---the result figures were missing. These issues would generate too much of confusion for the readers. Especially because of the missing of result figures made me can't judge the quality of the manuscript. Therefore, I can't recommend acceptance for this manuscript. My main concerns were on the rigorousness of the study.

1. First, no figures were attached with the recent manuscript. I can only find three figures in previous submission, which might not be the latest version.

2. Second, I only see three Insert Figure positions (two Figure 2 insert) with figure captions. But the author mentioned Fig 4 (at least once in line 422), and Fig 5 multiple times (e.g. line 448, 468). This is unacceptable.

3. The tables were also needed to be confirmed before submitting the manuscript. The footnote of table 2 indicated Table 1. In Table 3, the value format was different in the "Range, Average (Standard Deviation)" column, e.g. first three rows were different. And, what is "m"; is it necessary to write "m"?

4. The term "ground-truthing" is misleading. The process of ground-truthing is merely one step of their data collection process. As mentioned by the authors: "ground truthing involves driving to locations to visibly verify existing conditions of the built environment [36]". This is a common practice is many social and geographical studies. Based on the aims (as written in section 1.3) and results, the authors did not make any contribution to "ground-truthing".

5. Using previous submission figures. The figure quality (for Fig 1 and 2) is unacceptable to be published. Especially Fig 1, no word in the maps can be recognized except the three location names. Therefore, all things written in lines 254-261 can't be observed from the maps. In addition, the organization of the maps are also not suitable for publications.

6. In section 2.3, the authors described a survey way of data collection, which produced the primary data in this study. The authors should also explicitly mention how many "graduate research assistants" were participated in the data collection process. Moreover, because this is the main data in the study, the authors should also provide a full list of questions in appendix or in the article to tell the readers what data were collected. The only question mentioned by the author was the example of vandalism.

7. Figure 2 in the previous submission is problematic. It presented as a line plots with two lines flow across the horizontal axis, with horizontal axis showing different factors and vertical axis showing the risk of exposures. Line plots can only be used to show trends that have a numerical and continuous horizontal and vertical axes. In this figure, the horizontal axis presented factors, which orders do not have any specific meanings.

Reviewer #4: This study aims to analyse the correlation between environmental risk and built environmental design and conducts precise analyses to demonstrate how the environment risks correlated with the built environment and vulnerability. The paper is well-organised, but the main worries are about the representation of environment design.

First, since the authors clarify that the aim is to see if there was a double threat of environmental risk and poor built environmental design, but the current presentation about environment design is still weak. The authors talked more about environment design and society (health, justices), how to measure environment design, how to evaluate environmental risks in the literature review, but the relationship between design and environmental risks is less mentioned. For example, in existing findings, what kinds of characteristics of the built environment can exacerbate the environmental risks, and can POE evaluate these characteristics effectively, is it validate to apply POE to study risk-related environment design? Or else, how to explain the usage of variables in table 6?

Second, although the analyses from 3.2, 3.3 and 3.4 are helpful, the necessary explanations of variables in MLM are in a lack, and the results are less convincing, which need more evidence. For example, ¨suggesting that places without easy access by car, bus, walking, or biking have reduced environmental risk¨, which is biased. If people cannot walk and cycle to these places, what is a need to study these places? The demonstrations about the analysis and results of MLM need improvement.

Additionally, more discussions about the results of MLM are in need. For example, why and how the variables are related to environmental risk from the findings of this paper. In all, the presentation about the correlation between environmental risk and design should be investigated more.

Minor comments:

1) In the section of 2.5, the structure of analysis should be clarified more, such as the connections of PCA and MLM analyses.

2) Please pay attention to the length of the introduction and literature. The first two paragraphs in literature are repeatable with the introduction, it is better to make them concise.

3) Line 169-170, is this paper the first one to apply the instrument Cross and Küller? If not, it is in a need to mention how others use it in urban and environmental studies.

4) Line 339-340, to be clear about the vulnerability, is it about healthy vulnerability or social vulnerability?

5) Line 389, the subtitle ¨Physical and social condition¨, which is not clear, is physical health or physical environment, society or socioeconomic factors?

6) Complement more information about the data (e.g.: year of data, quality).

---

## [Author Response · Author response to Decision Letter 1]

2 Feb 2021

Rebuttal 

Revision 2

PONE-D-20-06092R1

Ground Truthing” Environmental Risk with Design: Translational Research using a Multi-Site, Multi-City Approach for Iowa’s Diversifying Small Towns

PLOS ONE

6. Review Comments to the Author

Reviewer #3: This manuscript presented a quantitative analysis for the understanding of equity of environment conditions and well being in small towns or rural areas in Iowa, USA. The topic is important and interesting. But, the manuscript is not ready for publication. The current status of the manuscript was poorly written, with too much of missing information, e.g. the data collection was not clear, how the data was analyzed from one step to another was not clearly described, and the most critical issue---the result figures were missing. These issues would generate too much of confusion for the readers. Especially because of the missing of result figures made me can't judge the quality of the manuscript. Therefore, I can't recommend acceptance for this manuscript. My main concerns were on the rigorousness of the study.

1. First, no figures were attached with the recent manuscript. I can only find three figures in previous submission, which might not be the latest version.

RESPONSE: all figures were uploaded through PLOS One’s direction and I have no idea why they weren’t available as part of this submission. Prior to this second revision being sent to reviewers, I had multiple emails with staff at PLOS One to ensure that none of the figures violated copyright. Referring to the above mentioned critical issue, the result figures were actually improved through the review process and reflect the intent of peer review to ensure rigor and clarity when presenting scientific research.

2. Second, I only see three Insert Figure positions (two Figure 2 insert) with figure captions. But the author mentioned Fig 4 (at least once in line 422), and Fig 5 multiple times (e.g. line 448, 468). This is unacceptable. 

RESPONSE Fixed the second Fig. 2 reference on line 427 to be Figure 4, and the second figure 3 reference on line 531 and 532 to figure 5. Insert fig 1 was at 252, insert figure 2 was at 358, insert figure 3 was at 374, insert figure 4 was at 427, and insert figure 5 was at 531. I double-checked the manuscript to verify that figures are now accurate and the correct ones are references within paragraphs. 

3. The tables were also needed to be confirmed before submitting the manuscript. The footnote of table 2 indicated Table 1. In Table 3, the value format was different in the "Range, Average (Standard Deviation)" column, e.g. first three rows were different. And, what is "m"; is it necessary to write "m"?

RESPONSE: All tables were corrected for labels and in-text reference as needed throughout the document. Cleaned up table 3. 

4. The term "ground-truthing" is misleading. The process of ground-truthing is merely one step of their data collection process. As mentioned by the authors: "ground truthing involves driving to locations to visibly verify existing conditions of the built environment [36]". This is a common practice is many social and geographical studies. Based on the aims (as written in section 1.3) and results, the authors did not make any contribution to "ground-truthing".

RESPONSE: This is the second revision of the manuscript and the second time a reviewer has made this comment. I’ve revised the title to eliminate “ground truthing” and replaced it with crosscutting, which better reflects the environmental justice nature of the paper. Similar revisions happened throughout the document. The suggestion is welcome and I hope the new title better reflects the intent of the manuscript. 

5. Using previous submission figures. The figure quality (for Fig 1 and 2) is unacceptable to be published. Especially Fig 1, no word in the maps can be recognized except the three location names. Therefore, all things written in lines 254-261 can't be observed from the maps. In addition, the organization of the maps are also not suitable for publications.

RESPONSE: Maps were updated following the previous submission. Nevertheless, I re-processed the maps for a higher resolution. Short of remaking the maps, which would require extensive time unwarranted in a revision, the maps clearly indicate transect locations and buffer overlaps, suitable for replication. Similarly, all figures were updated to higher resolution as part of the first revised manuscript. 

6. In section 2.3, the authors described a survey way of data collection, which produced the primary data in this study. The authors should also explicitly mention how many "graduate research assistants" were participated in the data collection process. Moreover, because this is the main data in the study, the authors should also provide a full list of questions in appendix or in the article to tell the readers what data were collected. The only question mentioned by the author was the example of vandalism.

RESPONSE: Added appendix 5 to the appendices showing an image of how survey data was collected in the field. No questions were asked of people, ever, as noted in the ethics statement. As noted in the manuscript, environmental quality variables are from the Cross and Kuller studies cited in the paper, and a complete list of survey questions is available from them upon request. I don’t have permission to publish the survey instrument. Added number of graduate students.

7. Figure 2 in the previous submission is problematic. It presented as a line plots with two lines flow across the horizontal axis, with horizontal axis showing different factors and vertical axis showing the risk of exposures. Line plots can only be used to show trends that have a numerical and continuous horizontal and vertical axes. In this figure, the horizontal axis presented factors, which orders do not have any specific meanings.

RESPONSE: Changed chart to scatterplot style and added categorical labels to the horizontal axis for reference (figure 4 now). 

Reviewer #4: This study aims to analyse the correlation between environmental risk and built environmental design and conducts precise analyses to demonstrate how the environment risks correlated with the built environment and vulnerability. The paper is well-organised, but the main worries are about the representation of environment design.

First, since the authors clarify that the aim is to see if there was a double threat of environmental risk and poor built environmental design, but the current presentation about environment design is still weak. The authors talked more about environment design and society (health, justices), how to measure environment design, how to evaluate environmental risks in the literature review, but the relationship between design and environmental risks is less mentioned. [RESPONSE: revised the manuscript throughout to reinforce why environmental design matters and how the study focuses on the intersection of environmental design and risk] For example, in existing findings, what kinds of characteristics of the built environment can exacerbate the environmental risks, and can POE evaluate these characteristics effectively, is it validate to apply POE to study risk-related environment design? Or else, how to explain the usage of variables in table 6?

Second, although the analyses from 3.2, 3.3 and 3.4 are helpful, the necessary explanations of variables in MLM are in a lack, and the results are less convincing, which need more evidence. For example, ¨suggesting that places without easy access by car, bus, walking, or biking have reduced environmental risk¨, which is biased. If people cannot walk and cycle to these places, what is a need to study these places?[RESPONSE: revised this section and provided more detail into the variables used for MLM] The demonstrations about the analysis and results of MLM need improvement. 

Additionally, more discussions about the results of MLM are in need. For example, why and how the variables are related to environmental risk from the findings of this paper. In all, the presentation about the correlation between environmental risk and design should be investigated more. [RESPONSE: this is discussed in the results and discussion, I would need to write another paper to go into further detail on this very important point.]

Minor comments: [Response: I don’t know if these are minor. It took quite a bit of time to resolve them, but thanks for the feedback.]

1) In the section of 2.5, the structure of analysis should be clarified more, such as the connections of PCA and MLM analyses. 

RESPONSE: added a couple sentence and revised the first paragraph to make these connections clear. 

2) Please pay attention to the length of the introduction and literature. The first two paragraphs in literature are repeatable with the introduction, it is better to make them concise.

RESPONSE: Cut the introduction down substantially and eliminated redundant text further.

3) Line 169-170, is this paper the first one to apply the instrument Cross and Küller? If not, it is in a need to mention how others use it in urban and environmental studies. RESPONSE: done and citations added.

4) Line 339-340, to be clear about the vulnerability, is it about healthy vulnerability or social vulnerability?

RESPONSE: revised to clearly indicate social vulnerability.

5) Line 389, the subtitle ¨Physical and social condition¨, which is not clear, is physical health or physical environment, society or socioeconomic factors? 

RESPONSE: Revised heading to emphasize that this is the environmental design criteria of physical and social conditions.

6) Complement more information about the data (e.g.: year of data, quality). RESPONSE: updated in appendices.

---

## [Decision Letter · Decision Letter 2]

22 Mar 2021

PONE-D-20-06092R2

Crosscutting Environmental Risk with Design: A Multi-Site, Multi-City Socioecological Approach for Iowa’s Diversifying Small Towns

PLOS ONE

Dear Dr. Shirtcliff,

Thank you for submitting your manuscript to PLOS ONE. After careful consideration, we feel that it has merit but does not fully meet PLOS ONE’s publication criteria as it currently stands. Therefore, we invite you to submit a revised version of the manuscript that addresses the points raised during the review process.

We look forward to receiving your revised manuscript.

Kind regards,

Tzai-Hung Wen, Ph.D.

Academic Editor

PLOS ONE

Journal Requirements:

Reviewers' comments:

Reviewer's Responses to Questions

**Comments to the Author**

1. If the authors have adequately addressed your comments raised in a previous round of review and you feel that this manuscript is now acceptable for publication, you may indicate that here to bypass the “Comments to the Author” section, enter your conflict of interest statement in the “Confidential to Editor” section, and submit your "Accept" recommendation.

Reviewer #3: (No Response)

Reviewer #4: All comments have been addressed

2. Is the manuscript technically sound, and do the data support the conclusions?

Reviewer #3: Yes

Reviewer #4: Yes

3. Has the statistical analysis been performed appropriately and rigorously? 

Reviewer #3: I Don't Know

Reviewer #4: Yes

4. Have the authors made all data underlying the findings in their manuscript fully available?

Reviewer #3: No

Reviewer #4: Yes

5. Is the manuscript presented in an intelligible fashion and written in standard English?

Reviewer #3: Yes

Reviewer #4: Yes

6. Review Comments to the Author

Reviewer #3: This manuscript presented a series of analyses including post occupancy evaluation (POE), principal component analysis (PCA), and multi-level model (MLM) to study the parallel communities in several small towns in Iowa, for the understanding of how built environment affect the quality of living and health risks to different people. This is the second time I review this manuscript, and all my previous concerns were addressed in current version. My new concerns are about the MLM analysis and interpretation which I couldn't review for, due to the incompleteness of previous version. There are also some minor issues that require further refine before publication.

1. Figure 2, the Y-axis and legend are missing. I can imagine blue bars are component 1 and orange bars are component 2, but please be explicit as this is a formal publication. In addition, the information in Figure 2 and Table 4 is exactly same, hence redundant.

2. Same as above for Figure 3. And Figure 3/Table 5.

3. For Figure 4, the estimated ‘beta’(s) are known as coefficient(s) or slopes, but the term coefficient is more preferred; and the main issue is that the term ‘beta’ is less formal and not preferred.

4. Second issue about Figure 4 is that in the figure, the Y-axis is written as ‘Increase in Environmental Risk’—how does this MLM slopes can be used to interpret as the ‘increase in environmental risk’? Please explain.

5. The process of running the MLM is not clear. E.g., what is(are) the dependent variable(s) in the ‘multiple multilinear models’ (page 18 line 391)? How does the ‘multiple multilinear model’ works? This line (391) is under section of ‘3.5 POE and MLM’, but no ‘multilinear model’ is described in analysis method section (sec 2.5). Based on the sec 2.5 lines 293-306, as a reader I would expect only one multivariate multilevel model will be presented.

6. Is it valid to compare different models? Related to previous point, it seems unclear, and based on what were written by the authors, there are more than one models in the study and presented in table 6/figure 4.

7. Table 6, I would suggest the authors to use ‘*’ to show the statistical significance instead of italicized. Mystery and complexity are not italicized in table 6, are them not significant? Access were italicized, but according to the table footnote, all but access are significant (p<.05). The description in footnote and written in table is conflict.

8. Please make sure all relevant data is provided. It seems like the data described in sec 2.3---those data collected with the Fulcrum tool--- is not provided by the authors. According to Plos One policy, all data underlying the findings should be provided without restriction. The data in Table 2 is only showing the mean, SD etc., and the data points behind this info should be provided. The raw data in sec 2.4 seems download-able from publicly available source so it should be fine, but it would be better if a cleaned and extracted copy for only the study sites can also be provided.

Reviewer #4: This paper addressed the comments well in the previous version and improved the quality of the paper. Overall, the paper is well-organized, but I still have a few minor comments that need authors to spend more time on that: 1) the introduction is still too long. For section1.1 and 1.2, there is a single paragraph at the beginning, but it is not necessary. These parts are also wordy. Also, the logistic between sections is confusing: why section 1.1 is the literature review? It will be apparent if the authors can make the introduction concise and straightforward. 2) The figures are expected in high-quality, especially for the results, but figure 2 and figure 3 are not qualified, missing the axis, title and legend. The texts in Figure 4 are too close and not clear to read. 3) The limitations of this paper are not mentioned enough, which also embodied in the data collection and qualitative method.

7. PLOS authors have the option to publish the peer review history of their article (what does this mean?). If published, this will include your full peer review and any attached files.

Reviewer #3: No

Reviewer #4: No

---

## [Author Response · Author response to Decision Letter 2]

11 Apr 2021

I have included a response to the reviewers in the attached revision 3 rebuttal letter. Thank you for your generous reviews to date. The manuscript has greatly benefited from this process in terms of rigor and clarity. I hope the revisions meet expectations.

---

## [Decision Letter · Decision Letter 3]

11 May 2021

Crosscutting Environmental Risk with Design: A Multi-Site, Multi-City Socioecological Approach for Iowa’s Diversifying Small Towns

PONE-D-20-06092R3

Dear Dr. Shirtcliff,

We’re pleased to inform you that your manuscript has been judged scientifically suitable for publication and will be formally accepted for publication once it meets all outstanding technical requirements.

Kind regards,

Tzai-Hung Wen, Ph.D.

Academic Editor

PLOS ONE

Additional Editor Comments (optional):

Reviewers' comments:

Reviewer's Responses to Questions

**Comments to the Author**

1. If the authors have adequately addressed your comments raised in a previous round of review and you feel that this manuscript is now acceptable for publication, you may indicate that here to bypass the “Comments to the Author” section, enter your conflict of interest statement in the “Confidential to Editor” section, and submit your "Accept" recommendation.

Reviewer #3: All comments have been addressed

Reviewer #4: All comments have been addressed

2. Is the manuscript technically sound, and do the data support the conclusions?

Reviewer #3: Yes

Reviewer #4: Yes

3. Has the statistical analysis been performed appropriately and rigorously? 

Reviewer #3: Yes

Reviewer #4: Yes

4. Have the authors made all data underlying the findings in their manuscript fully available?

Reviewer #3: Yes

Reviewer #4: Yes

5. Is the manuscript presented in an intelligible fashion and written in standard English?

Reviewer #3: Yes

Reviewer #4: Yes

6. Review Comments to the Author

Reviewer #3: (No Response)

Reviewer #4: I have no specific comments with regard to the current version. The paper has been improved and meet the requirement after addressing.

---

## [Editor Report · Acceptance letter]

11 Jun 2021

PONE-D-20-06092R3 

Crosscutting Environmental Risk with Design: A Multi-Site, Multi-City Socioecological Approach for Iowa’s Diversifying Small Towns 

Dear Dr. Shirtcliff:

I'm pleased to inform you that your manuscript has been deemed suitable for publication in PLOS ONE. Congratulations! Your manuscript is now with our production department. 

Kind regards, 

on behalf of

Dr. Tzai-Hung Wen 

Academic Editor

PLOS ONE